# Characterization of Oxide Film of Implantable Metals by Electrochemical Impedance Spectroscopy

**DOI:** 10.3390/ma12213466

**Published:** 2019-10-23

**Authors:** Yoshimitsu Okazaki

**Affiliations:** National Institute of Advanced Industrial Science and Technology, 1-1-1 Higashi, Tsukuba-shi, Ibaraki 305-8566, Japan; y-okazaki@aist.go.jp

**Keywords:** Ti–15Zr–4Nb alloy, implantable metals, oxide film resistance, capacitance, electrochemical impedance spectroscopy

## Abstract

The oxide film resistance (R_P_) and capacitance (C_CPE_) diagrams of implantable metals (commercially pure Ti, four types of Ti alloys, Co–28Cr–6Mo alloy, and stainless steel) were investigated by electrochemical impedance spectroscopy (EIS). The thin oxide film formed on each implantable metal surface was observed in situ by field-emission transmission electron microscopy (FE-TEM). The Ti–15Zr–4Nb–1Ta and Ti–15Zr–4Nb–4Ta alloys had higher oxygen concentrations in the oxide films than the Ti–6Al–4V alloy. The thickness (d) of the TiO_2_ oxide films increased from approximately 3.5 to 7 nm with increasing anodic polarization potential from the open-circuit potential to a maximum of 0.5 V vs. a saturated calomel electrode (SCE) in 0.9% NaCl and Eagle’s minimum essential medium. R_P_ for the Ti–15Zr–4Nb–1Ta and Ti–15Zr–4Nb–4Ta alloys was proportional to d obtained by FE-TEM. C_CPE_ was proportional to 1/d. R_P_ tended to decrease with increasing C_CPE_. R_P_ was large (maximum: 13 MΩ·cm^2^) and C_CPE_ was small (minimum: 12 μF·cm^−2^·s^n−1^, n = 0.94) for the Ti–15Zr–4Nb–(0 to 4)Ta alloys. The relative dielectric constant (ε_r_) and resistivity (k_OX_) of the oxide films formed on these alloys were 136 and 2.4 × 10^6^–1.8 × 10^7^ (MΩ·cm), respectively. The Ta-free Ti–15Zr–4Nb alloy is expected to be employed as an implantable material for long-term use.

## 1. Introduction

Stainless steels, Co–Cr–Mo alloys, commercially pure Ti (C.P.-Ti), and Ti alloys have been widely used for orthopedic implants. The mechanical strength of stainless steel is increased by adding nitrogen (N) and 20% cold working [1]. C.P.-Ti is classified into grade (G)-1, G-2, G-3, and G-4. In particular, the mechanical strength of C.P. G-4 Ti is also increased by 20% cold working [2,3]. Among the Ti alloys, Ti–6Al–4V alloy (hereafter, alloy compositions are expressed in mass%) is widely used. Recently, low-cost manufacturing processes for highly biocompatible Ti–15Zr–4Nb–4Ta, Ti–15Zr–4Nb–1Ta, and Ta-free Ti–15Zr–4Nb alloys have been developed [4]. 

The static immersion and anode polarization tests of metallic biomaterials are standardized in JIS T 0304 and JIS T 0302, respectively [5,6]. As the static immersion test, a seven-day (7-d) immersion test is recommended. NaCl (0.9%) and Ringer’s solution are the recommended solutions for the corrosion test in ISO 16428 [7] and ISO 16429 [8]. NaCl (0.9%) solution adjusted to pH 2 by adding HCl also meets the requirements in ISO 16428 for use as a solution for severe testing. 

Owing to the recent rapid progress of transmission electron microscopy, the chemical composition and thickness of oxide films can be analyzed in situ by field-emission transmission electron microscopy (FE-TEM) [9]. We focused on the analysis by FE-TEM of the oxide film thickness (d) and the change in the chemical composition of oxide films formed on Ti–15Zr–4Nb–4Ta, Ti–15Zr–4Nb–1Ta, Ti–6Al–4V, and Co–28Cr–6Mo alloys and high-N stainless steel. 

Electrochemical impedance spectroscopy (EIS) is a useful technique to determine electrochemical reactions at a metal/oxide film interface [10,11,12,13,14,15,16,17,18,19,20]. In a surface oxide film, the capacitance (C, μF in this work) is inversely proportional to the oxide film thickness (d) in nanometers (1 nm = 10^−7^ cm) and proportional to the electrode area (A) exposed to a solution in square centimeters (1 cm^2^ in this work): C = ε_o_·ε_r_·A/d,(1)
where ε_o_ is the vacuum permittivity (8.854 × 10^−8^ μF/cm) [9] and ε_r_ is the relative dielectric constant of the oxide film [11,18,19,20]. C is proportional to the reciprocal of the thickness (1/d). Also, the oxide film resistance (R_P_, MΩ in this work) is expressed by:R_P_ = k_OX_·d/A,(2)
where k_OX_ is the resistivity (MΩ·cm) of the oxide film [21,22]. R_P_ is proportional to d and inversely proportional to A. As it is difficult to directly measure the thickness (d) of an oxide film in solution, we estimated ε_r_ and k_OX_ for the oxide films of metal surfaces using d measured by FE-TEM.

From Equations (1) and (2), Equation (3) is obtained.

R_P_ = ε_o_·ε_r_·k_OX_ /C(3)

Thus, R_P_ and C are inversely proportional. There have been no reports on the systematic investigation of R_P_ and C diagrams for implantable metals. In this study, we focused on ε_r_, k_OX_, and R_P_ and C diagrams to estimate the electrochemical stability of passive oxide films formed on various implantable metals by anodic polarization using Equation (3). The effects of biological solutions on the R_P_ and C diagrams were also examined.

The purpose of this study was to establish R_P_ and C diagrams of implantable metals and, in particular, to estimate the wide ranges of R_P_ and C values for Ti materials, stainless steel, and Co–Cr–Mo alloy. Another aim was to obtain the electrochemical data of typical implantable metals under the same conditions. To estimate ε_r_ and k_ox_ and obtain R_P_ and C diagrams for implantable metals, we investigated—by FE-TEM and EIS–the electrochemical stability of oxide films formed on Ti–15Zr–4Nb–4Ta, Ti–15Zr–4Nb–1Ta, Ti–15Zr–4Nb, and Ti–6Al–4V alloys, C.P.-Ti G-4, high-N stainless steel, and Co–28Cr–6Mo alloy. The oxide films on these metals were mainly formed in an anodic polarization or 7-d immersion test in 0.9% NaCl and Eagle’s minimum essential medium (MEM) at 37 °C. The chemical composition and thickness of oxide films formed on the above materials after anodic polarization were analyzed in situ by FE-TEM. R_P_ and C for each oxide film were measured by EIS and were compared among the various implantable metals using a single-oxide-layer model to fit the EIS data. We estimated ε_r_ and k_ox_ using d obtained by FE-TEM. 

Moreover, we examined the relationship between R_P_ and C, as well as the effects of materials on R_P_ and C obtained in 0.9% NaCl solution at 37 °C. Moreover, the effects of biological solutions on these values were examined using 0.9% NaCl; Eagle’s MEM (Nissui) without fetal bovine serum (FBS; FBS-free Eagle’s MEM); Eagle’s MEM containing 10 vol% FBS and 7.5 vol% NaHCO_3_; Ringer’s solution, 0.9% NaCl containing 25 vol% calf serum; and 0.9% NaCl containing HCl (0.9% NaCl + HCl, pH = 1) [23]. The R_P_ and C diagrams obtained in this study are useful for the development of new metallic materials with excellent corrosion resistance.

## 2. Materials and Methods 

### 2.1. Materials

Vacuum arc melting was performed on biocompatible Ti–15Zr–4Nb–4Ta, Ti–15Zr–4Nb–1Ta, and Ta-free Ti–15Zr–4Nb alloys [4]. After β-forging, α-β forging was conducted to obtain the α- and β-structures. Rods of 25 mm diameter were manufactured by α–β forging, after which the rods were annealed at 700 °C for 2 h [2,4]. For comparison, four types of alloy (Ti–6Al–4V, C.P.-Ti G-4, Co–28Cr–6Mo, and high-N stainless steel) were prepared. The Ti–6Al–4V rods were annealed at 700 °C for 2 h [4]. The Co–28Cr–6Mo rods were annealed at 1200 °C for 1 h [24]. The high-N stainless steel rods were solution-treated at 1150 °C for 1 h and then quenched in water (solution-treated). Part of the C.P.-Ti G-4 rods were 20% cold-worked (20% CW) after annealing [3], and the high-N stainless steel billets were 20% cold-worked after solution treatment [25]. Table 1 shows the chemical compositions (in mass%) of the four Ti alloys and the other metals used for comparison.

### 2.2. FE-TEM Observation of Oxide Film 

Ti–15Zr–4Nb–4Ta, Ti–15Zr–4Nb–1Ta, and Ti–6Al–4V alloys, annealed C.P.-Ti G-4, solution-treated high-N stainless steel, and annealed Co–28Cr–6Mo alloy specimens were polished with 1000, 1200, 2400, and 4000 grit waterproof emery papers, and then mirror-like-finished by buff cleaning with OP-S or Al_2_O_3_ (for stainless steel) suspension. After mirror-like finishing, an oxide film was formed on the alloy surface by anodic polarization from the open-circuit potential to a maximum of 0.5 V vs. a saturated calomel electrode (SCE) at a sweep rate of 40 mV/min in 0.9% NaCl and Eagle’s MEM containing FBS, or in a 7-d static immersion test in 0.9% NaCl at 37 °C [9]. The temperature of the test solutions was controlled within the range of 37 ± 1 °C. Anodic polarization and 7-d static immersion tests were conducted in accordance with JIS T 0302 [6] and JIS T 0304 [5]. In this study, because the potential at which the passive film fully formed on metal surfaces was stable, we selected a maximum of 0.5 V vs. SCE for FE-TEM measurement.

To prepare FE-TEM samples, we used FB-2000A focused ion beam (FIB) processing (Hitachi, Ltd.) and microsampling systems. Each alloy sample (10 × 15 × 3 µm) was observed using an JEM–2010F system (JEOL, Ltd. Akishima, Tokyo, Japan) at an acceleration voltage of 200 kV and magnifications of 10,000 × to 1,000,000 ×. Sigma energy-dispersive X-ray spectrometry (EDX, Vantage, Akishima, Tokyo, Japan) was carried out to analyze the composition of the oxide film. The concentration of each element in the oxide film was measured by EDX. The inside of the thin oxide film was measured obliquely so that the EDX spots did not overlap. In this study, we assumed that the oxide film formed on the metal surface in the solution had a constant thickness in vacuum in FE-TEM measurement.

### 2.3. Electrochemical Model of Oxide Film 

The electrochemical model used to analyze the EIS data is shown in Figure 1a [9,10,12]. Figure 1b shows the equivalent circuit model used to analyze the EIS data [9,10,12,14,15,17,18,19,20,21,26]. Figure 1c shows a schematic illustration of the Nyquist plot for the equivalent circuit. The following equation holds for this circuit:(4)$${(x-\frac{2R_{S}+R_{P}}{2})}^{2}+y^{2}={\left(\frac{R_{P}}{2}\right)}^{2}$$
where y and x respectively represent the imaginary and real components of the complex impedance (Ω·cm^2^) [9]; R_S_ and R_P_ are the solution and film resistances, respectively; and C_CPE_ is the constant-phase element, which includes the capacitance behavior of the surface film. The impedance representation of the constant-phase element was given by Z_CPE_ = 1/[(jω)^n^ ·Q], since the actual measurement produces an incomplete circle on a Nyquist diagram [9,11,13,15,18,19,20,21,27]. Q (unit, μF/cm^2^·s^n−1^) is a capacitance-like element, j is the imaginary unit, ω (rad/s) = 2πf (f: frequency) is the angular frequency, and n is an exponent between 0 and 1 (when n = 1, a system behaves as an ideal capacitor, and when n = 0, it behaves as an ideal resistor). That is, when n is close to 1, Q is close to the ideal capacitance C [9,11,13,15,17,20,27]. Q was estimated using HZ-5000ANAEIS-Analysis analytical software (Ver. 1.0.0, Hokuto Denkou, Meguro-ku, Tokyo, Japan) after the impedance test. Q is represented by C_CPE_ in this work [11]. R_P_ and R_S_ are calculated by fitting the measurement data with analytical software, and C_CPE_ is determined from the equation ω_top_ = 1/(C_CPE_·R_P_).

### 2.4. EIS Measurements

Ti–15Zr–4Nb–4Ta, Ti–15Zr–4Nb–1Ta, Ti–15Zr–4Nb, Ti–6Al–4V, C.P.-Ti G-4 (annealed and 20% CW), Co–28Cr–6Mo, and high-N stainless steel (solution-treated and 20% CW) were used for EIS measurements. Specimens of 1 cm^2^ surface area (for normalization) were soldered with lead for electrical connection. Thereafter, the samples were coated with epoxy resin. Then, their surfaces were finished using mainly 1000-grit waterproof emery paper (1000-grit-polished) and ultrasonically cleaned in ethanol for 5 min in accordance with JIS T 0302 [6]. To examine the effect of mirror-like finishing of the specimens used for FE-TEM analysis on EIS properties, some of the specimens were buff-cleaned using OP-S (for Ti materials and Co–28Cr–6Mo alloy) or Al_2_O_3_ (for stainless steel) suspension. Electrochemical impedance was measured independently in 0.9% NaCl (pH = 6.0), FBS-free Eagle’s MEM (pH = 6.4), 10 vol% FBS-containing Eagle’s MEM (pH = 8.3), 0.9% NaCl containing 25 vol% calf serum (pH = 6.8), Ringer’s solution (pH = 7.2, NaCl 8.36 g/L, CaCl_2_ 0.15 g/L, KCl 0.3 g/L), and 0.9% NaCl containing HCl (pH = 1). 

An HZ-5000 electrochemical measurement system (Hokuto Denkou, Meguro-ku, Tokyo, Japan) with a potentiostat and a frequency response analyzer (FRA) was used for the EIS measurement, as shown in Figure 2. Nyquist and Bode plots and n values were measured using HZ-5000ANAEIS-Analysis analytical software (Ver. 1.0.0, Hokuto Denkou, Meguro-ku, Tokyo, Japan). The experimental cell shown in Figure 2 was used [9]. The reference electrode (R.E.) was electrically connected to a Luggin capillary via an agar salt bridge. The distance between the sample surface and the tip of the Luggin capillary was set to 5 mm. The temperature of the test solution was held at 37 ± 1 °C. The alternating current was measured at a constant alternating potential of 10 mV, while the frequency was changed from 100 kHz to 10 mHz to calculate the real and imaginary components, the absolute value, and the phase difference of the impedance. The measurements were repeated three times. To investigate the effects of the oxide film thickness on the impedance properties, an anodic polarization or 7-d static immersion test was conducted before the impedance test. The anodic polarization test was performed at a sweep rate of 40 mV/min starting from the open-circuit potential to a maximum of 1 V vs. SCE. The 7-d static immersion test of the Ti–15Zr–4Nb–1Ta and Ti–6Al–4V alloys was performed in 0.9% NaCl, in which a beaker containing test samples was placed inside an incubator at 37 ± 1 °C for 7 days.

## 3. Results and Discussion

### 3.1. FE-TEM Analysis of Oxide Films 

We observed the thin oxide films formed on the implantable metals in situ. Figure 3 shows FE-TEM images of the oxide films formed on the Ti–15Zr–4Nb–1Ta, Ti–6Al–4V, Co–28Cr–6Mo, and solution-treated high-N stainless steel surfaces by anodic polarization up to 0 V vs. SCE in 0.9% NaCl at 37 °C. To compare the distribution of each metallic element in the oxide films formed on metal surfaces, we selected 0 V vs. SCE as the potential in the passive region. As can be seen from the FE-TEM images, the surface of each oxide film is protected by carbon, and a passive oxide film can be observed on the metal surface. According to the electron diffraction patterns for the Ti–15Zr–4Nb–1Ta and Ti–6Al–4V alloys, the metals had an hcp (hexagonal–close–packed) structure, whereas the oxide films had an amorphous structure. Thin oxide films of 4.2 ± 0.2, 3.2 ± 0.2, 1.7 ± 0.2, and 2.3 ± 0.1 nm thicknesses (mean ± standard deviation) were observed on the Ti–15Zr–4Nb–1Ta, Ti–6Al–4V, Co–28Cr–6Mo, and high-N stainless steel surfaces after anodic polarization up to 0 V vs. SCE, respectively. Figure 4 shows EDX spectra of the oxide films at the positions shown in Figure 3. The small Cu, Ga, and Mo peaks originate from the microsampling holder. Peaks of oxygen and each metal element constituting the oxide film were observed. The oxide film that formed on Ti–15Zr–4Nb–1Ta alloy consisted of TiO_2_ containing Zr and small amounts of Nb and Ta, as shown in Figure 4a. The oxide film on Ti–6Al–4V alloy consisted of TiO_2_ containing Al and a small amount of V. The oxide film on Co–28Cr–6Mo alloy consisted of Cr_2_O_3_ containing Co and a small amount of Mo. The oxide film on high-N stainless steel consisted of Cr_2_O_3_ containing Fe and a small amount of Ni, as shown in Figure 4d.

Figure 5 shows the changes in the concentration (at%) of each element in the oxide films from the oxide/metal interface to the oxide film surface obtained by the EDX analysis of the oxide films. At the oxide/metal interface, the metal concentration was relatively high, and with increasing distance from the oxide/metal interface, the metal concentration decreased and the oxygen concentration increased. It was found that the Ti–15Zr–4Nb–1Ta alloy had a higher oxygen concentration in the oxide film than the Ti–6Al–4V alloy. In the oxide films formed on the Co–28Cr–6Mo alloy and high-N stainless steel surfaces, the concentrations of Co and Fe at the oxide/metal interface were high and decreased as the distance from the oxide/metal interface increased. The changes in oxygen concentration in the oxide films obtained by EDX analysis among the Ti alloys (Ti–15Zr–4Nb–4Ta, Ti–15Zr–4Nb–1Ta, Ti–6Al–4V, and C.P.-Ti G-4) are shown in Figure 6. The results of the 7-d immersion tests of Ti–15Zr–4Nb–4Ta and Ti–15Zr–4Nb–1Ta alloys are also shown for comparison with those of the anodic polarization tests up to 0 V vs. SCE. The oxygen concentration (at%) in the TiO_2_ oxide films increased in the order of C.P.-Ti G-4 (annealed) < Ti–6Al–4V < Ti–15Zr–4Nb–4Ta and Ti–15Zr–4Nb–1Ta alloys. Also, the change in oxygen concentration in the oxide films formed on the Ti–15Zr–4Nb–4Ta and Ti–15Zr–4Nb–1Ta alloys by anodic polarization up to 0 V vs. SCE was similar to that obtained in the 7-d immersion test. It is considered that this high oxygen concentration in the oxide films was due to the formation of ZrO_2_, Nb_2_O_5_, and Ta_2_O_5_ from Zr, Nb, and Ta, respectively. 

The effect of the anodic polarization potential on the thicknesses of the oxide films formed on the Ti–15Zr–4Nb–4Ta, Ti–15Zr–4Nb–1Ta, Ti–6Al–4V, and C.P.-Ti G-4 surfaces is shown in Figure 7. The thicknesses of the TiO_2_ oxide films after mirror-like finishing by buff cleaning with OP-S suspension were 3.8 ± 0.2 nm, as shown in Figure 7. For the Ti–15Zr–4Nb–4Ta and Ti–15Zr–4Nb–1Ta alloys, the thickness of each TiO_2_ oxide film increased from approximately 3.5 to 7 nm with increasing anodic polarization potential from the open-circuit potential to a maximum of 0.5 V vs. SCE in 0.9% NaCl and Eagle’s MEM. The thickness of the oxide film formed on the Ti–6Al–4V alloy surface in 0.9% NaCl tended to be slightly lower than those of the oxide films formed on the surfaces of the Ti–15Zr–4Nb–4Ta and Ti–15Zr–4Nb–1Ta alloys. The thickness of the oxide film formed on the C.P.-Ti G-4 surface after anodic polarization at 0 V vs. SCE in 0.9% NaCl at 37 °C was 4.7 ± 0.1 nm. Note that d = do + α (E_f_ − E_o_), where do is the thickness of the spontaneously grown oxide film, the slope α is the anodization coefficient (2.3 nm/V for the Ti–15Zr–4Nb–4Ta and Ti–15Zr–4Nb–1Ta alloys, coefficient of determination, R^2^ = 0.76), Eo is the open-circuit potential, and E_f_ is the potential at the end of anodic polarization. The α value was obtained by linear regression analysis. The anodization rates under the relevant TiO_2_ growth conditions have been reported to lie in the range of 2.5–3.25 nm/V [28].

### 3.2. EIS Analysis of Oxide Films 

Figure 8 shows Nyquist and Bode plots of oxide films on the Ti–15Zr–4Nb–1Ta, Ti–6Al–4V, Co–28Cr–6Mo, and solution-treated high-N stainless steel surfaces after anodic polarization up to 0 V vs. SCE in 0.9% NaCl at 37 °C. It can be seen that the typical implantable alloy/solution interface systems exhibit capacitive behavior. In the Bode plots shown in Figure 8b, |Z| tends to become constant at high frequencies, with the phase shift decreasing towards zero with increasing frequency. Highly capacitive behavior at low-to-medium frequencies is indicated by the phase shift approaching −90°, suggesting that a highly stable oxide film was formed on all the Ti–15Zr–4Nb–1Ta, Ti–6Al–4V, Co–28Cr–6Mo, and solution-treated high-N stainless steel surfaces. In this broad low-to-medium frequency range, the spectra had a constant slope of about −1. This is the characteristic response of an oxide film exhibiting capacitive behavior [29]. 

Figure 9a shows a plot of the oxide film resistance (R_P_) of the Ti–15Zr–4Nb–4Ta, Ti–15Zr–4Nb–1Ta, and Ti–6Al–4V alloys and C.P.-Ti G-4 versus the oxide film thickness (d) obtained by FE-TEM analysis after the anodic polarization tests from the open-circuit potential to a maximum of 0.5 V vs. SCE in 0.9% NaCl and Eagle’s MEM at 37 °C. For comparison, the results of the 7-d immersion test using the same Ti–15Zr–4Nb–4Ta and Ti–15Zr–4Nb–1Ta alloys in 0.9% NaCl at 37 °C are also shown in Figure 9a,b. The same marks in Figure 9a,b indicate the results for the same specimen. R_P_ was approximately proportional to d, as indicated by Equation (2). The obtained slope of the line for the Ti–15Zr–4Nb–4Ta and Ti–15Zr–4Nb–1Ta alloys was calculated to be 2.4 × 10^7^ MΩ·cm. The slope was determined by linear regression and R^2^ was 0.65. The k_OX_ values for the oxide films formed on the Ti–15Zr–4Nb–4Ta and Ti–15Zr–4Nb–1Ta alloys by anodic polarization up to 0 V or 0.5 V vs. SCE in 0.9% NaCl or Eagle’s MEM at 37 °C were calculated to be in the range of 2.4 × 10^6^–1.8 × 10^7^ MΩ·cm using Equation (2). k_OX_ for a rutile TiO_2_ single crystal was approximately 10^6^ MΩ·cm [30]. k_OX_ for the oxide film formed on C.P.-Ti G-4 by anodic polarization up to 0 V vs. SCE in 0.9% NaCl at 37 °C was calculated to be 2.48 × 10^6^ MΩ·cm. The resistivity of the oxide film formed on C.P.-Ti G-4 in phosphate-buffered saline (PBS) solution was 2.5 × 10^6^ MΩ·cm [22]. Both k_OX_ values for C.P.-Ti G-4 were nearly the same. As clearly shown in Figure 9a, R_P_ did not lie on a straight line passing through the origin, which is different from the result indicated in Equation (2), and R^2^ (0.65) of the slope was lower. This may have been caused by the effect of the thin film that initially formed on the metal surface during mirror-like polishing. We would like to investigate the cause of this difference in the future and improve the electrochemical model of the oxide film.

The unit of C_CPE_ was μF/cm^2^·s^n−1^. The n values obtained in this work were 0.86–1.0. In particular, the values of 0.91–0.96 were close to 1, indicating that the system behaved similarly to a capacitor. The C_CPE_ values obtained in this study were compared with those reported in the literature. Figure 9b shows a plot of C_CPE_ in the above unit for each oxide film versus the inverse of the oxide film thickness (1/d) measured by FE-TEM. C_CPE_ was proportional to 1/d, as indicated by Equation (1). The slope of the line represents ε_o_·ε_r_ for the thin TiO_2_ film. Expressing d in nanometers, C_CPE_ in microfarads, and A in square centimeters, ε_o_·ε_r_ for the Ti–15Zr–4Nb–4Ta and Ti–15Zr–4Nb–1Ta alloys was 1.20 × 10^−5^ μF/cm and ε_r_ was 136. The slope was determined by linear regression assuming a line through the origin, and R^2^ was 0.83. ε_r_ for the rutile TiO_2_ single crystal ranges from 89 to 173 in the literature [31]. The value of ε_r_ obtained in this study was within this range. From these results, we considered that the thickness of the oxide film formed on each metal surface in the solution was close to the value measured by FE-TEM.

R_P_ and C_CPE_ for the passive oxide films formed on the Co–28Cr–6Mo and solution-treated high-N stainless steel surfaces after anodic polarization up to 0 V vs. SCE in 0.9% NaCl at 37 °C were 0.42 ± 0.01 MΩ·cm^2^ and 33.0 ± 1.36 μF·cm^−2^·s^n−1^ (n = 0.92 ± 0.01) for Co–28Cr–6Mo, and 0.12 ± 0.01 MΩ·cm^2^ and 62.44 ± 1.19 μF·cm^−2^·s^n−1^ (n = 0.90 ± 0.002) for the solution-treated high-N stainless steel, respectively. To compare k_OX_ and ε_r_ for the Ti–15Zr–4Nb–4Ta and Ti–15Zr–4Nb–1Ta alloys, k_OX_ and ε_r_ for the oxide films formed on Co–28Cr–6Mo and high-N stainless steel by anodic polarization up to 0 V vs. SCE were calculated from the film thickness. The values for Co–28Cr–6Mo were 2.45 × 10^6^ MΩ·cm and 63, and the values for the solution-treated high-N stainless steel were 5.4 × 10^5^ MΩ·cm and 162, respectively. ε_r_ for the Co–28Cr–6Mo alloy was much higher than those (9.2 to 13.3) for Cr_2_O_3_ reported in the literature [26].

Comparing the results obtained after mirror-like polishing and 1000-grit polishing, R_P_ obtained for the mirror-like-finished specimens tended to be slightly higher than that for the 1000-grit-polished specimens. However, since the difference between them was small, we mainly measured the 1000-grit-polished specimens for convenience. Figure 10 shows the changes in R_P_ and C_CPE_ for the oxide films formed on the Ti–15Zr–4Nb–1Ta and Ti–6Al–4V alloys as a function of immersion time. R_P_ increased markedly up to an immersion time of 1 day and then saturated. Considering the results of R_P_ in the 7-d immersion test, we chose 0 V vs. SCE as the typical anodic oxidation potential. Figure 11 shows the changes in (a) R_P_ and (b) C_CPE_ for various implantable materials measured after anodic polarization at various potentials up to a maximum of 1 V vs. SCE in 0.9% NaCl at 37 °C as a function of anodic potential. The R_P_ values for the Ti–15Zr–4Nb–4Ta, Ti–15Zr–4Nb–1Ta, and Ti–15Zr–4Nb alloys, Ti–6Al–4V, and annealed Ti G-4 markedly increased with increasing anodic potential. The slope of the curve also changed near 0.5 V vs. SCE. On the other hand, the changes in the values for Co–28Cr–6Mo and solution-treated high-N stainless steel were extremely small compared with those for the Ti-based alloys. C_CPE_ was highest for the high-N stainless steel and lowest for the Ti–15Zr–4Nb–4Ta, Ti–15Zr–4Nb–1Ta, and Ti–15Zr–4Nb alloys among the seven materials. C_CPE_ increased with the degradation of the oxide film (0.8 V vs. SCE for Co–28Cr–6Mo), as shown in Figure 11b. From these results, we selected 0.5 vs. SCE as the typical anodic oxidation potential in the passive region.

Figure 12 shows the relationship between R_P_ and C_CPE_ (R_P_ and C_CPE_ diagram) obtained from the anodic polarization test using various metals (1000-grit-polished) up to a maximum of 1 V vs. SCE in 0.9% NaCl at 37 °C. To examine in detail the R_P_ and C_CPE_ diagrams in a wide range, anodic polarization was performed from the open-circuit potential up to a maximum of 1 V vs. SCE using nine metals. It was found that R_P_ and C_CPE_ were in inverse proportion, as indicated by Equation (3). For the high-N stainless steel and Co–28Cr–6Mo alloy, these values were concentrated on the side with large C_CPE_ (maximum: 80 μF·cm^−2^·s^n−1^, n = 0.90) and small R_P_ (minimum: 0.1 Ω·cm^2^) values. In contrast, for the Ti materials, they were concentrated on the side with small C_CPE_ and large R_P_ values. In particular, R_P_ and C_CPE_ for the Ti–15Zr–4Nb–4Ta, Ti–15Zr–4Nb–1Ta, and Ta-free Ti–15Zr–4Nb alloys had large R_P_ (maximum: 13 MΩ·cm^2^) and small C_CPE_ (minimum: 12 μF·cm^−2^·s^n−1^, n = 0.94) values. The effect of the 20% cold working of the high-N stainless steel and C.P.-Ti G-4 on R_P_ and C_CPE_ was also small.

The effects of biological solutions and pH on R_P_ and C_CPE_ for oxide films formed on Ti–15Zr–4Nb–4Ta alloy are shown in Figure 13. R_P_ and C_CPE_ were in inverse proportion in various solutions. Since the results obtained with various solutions approximately lie on a single curve, the effects of the biological solutions and pH on R_P_ and C_CPE_ were small. The effects of mirror-like finishing and 1000-grit polishing of the Ti–15Zr–4Nb–4Ta alloy immersed in Eagle’s MEM are also shown in Figure 13. It can be seen that the effects are small.

It is clear that the implantable metals had a highly capacitive property, namely, a highly stable oxide film was formed on their surfaces. For a metallic material with higher R_P_, the quantity of ions that pass through the surface oxide film is reduced, and such a material exhibits higher corrosion resistance. The R_P_ and C_CPE_ diagrams obtained in this study are useful for the development of new metallic materials with excellent corrosion resistance. The Ta-free Ti–15Zr–4Nb alloy is expected to be employed as an implantable material for long-term use in the human body.

## 4. Conclusions

We examined the electrochemical stability of oxide films formed on highly biocompatible Ti–15Zr–4Nb–4Ta, Ti–15Zr–4Nb–1Ta, and Ta-free Ti–15Zr–4Nb alloys and other implantable metals by FE-TEM and EIS. Thin oxide films of approximately 4.2 ± 0.2, 3.2 ± 0.2, 1.7 ± 0.2, and 2.3 ± 0.1 nm thicknesses were observed on Ti–15Zr–4Nb–1Ta, Ti–6Al–4V, Co–28Cr–6Mo, and high-N stainless steel surfaces after anodic polarization up to 0 V vs. SCE, respectively. By the EDX analysis of the thin oxide films, we found the metal concentration to be relatively high at the oxide/metal interface, and as the distance from the oxide/metal interface increased, the metal concentration decreased and the oxygen concentration increased. It was found that Ti–15Zr–4Nb–4Ta and Ti–15Zr–4Nb–1Ta alloys had a higher oxygen concentration in the oxide film than Ti–6Al–4V alloy. The oxide film formed on the Ti–15Zr–4Nb–1Ta alloy consisted of TiO_2_ with Zr and small amounts of Nb and Ta. The thickness of the TiO_2_ oxide film increased from 3.5 to 7 nm with increasing anodic polarization potential from the open-circuit potential to 0.5 V vs. SCE in 0.9% NaCl and Eagle’s MEM. The resistance (R_P_) of the oxide film on the Ti–15Zr–4Nb–4Ta, Ti–15Zr–4Nb–1Ta, and Ti–6Al–4V alloys was proportional to the oxide film thickness (d) obtained by FE-TEM. The capacitance (C_CPE_) of the oxide films was inversely proportional to d. R_P_ for the Ti–15Zr–4Nb–4Ta, Ti–15Zr–4Nb–1Ta, and Ta-free Ti–15Zr–4Nb alloys, Ti–6Al–4V, and C.P.-Ti G-4 markedly increased with increasing anodic potential in 0.9% NaCl at 37 °C. On the other hand, the changes in these values for Co–28Cr–6Mo and high-N stainless steel were extremely small compared with those for the Ti materials. R_P_ and C_CPE_ for the oxide films formed on various implantable metals showed an inversely proportional relationship, and R_P_ and C_CPE_ for the Ti–15Zr–4Nb–4Ta, Ti–15Zr–4Nb–1Ta, and Ta-free Ti–15Zr–4Nb alloys had a maximum value of 13 MΩ·cm^2^ and a minimum value of 12 μF·cm^−2^·s^n−1^ (n = 0.94). The relative dielectric constant and resistivity of oxide films formed on Ti–15Zr–4Nb–4Ta and Ti–15Zr–4Nb–1Ta alloys were 136 and 2.4 × 10^6^–1.8 × 10^7^ (MΩ·cm), respectively. The Ta-free Ti–15Zr–4Nb alloy is expected to be employed as an implantable material for long-term use in the human body.

## Figures and Tables

**Figure 1 materials-12-03466-f001:**
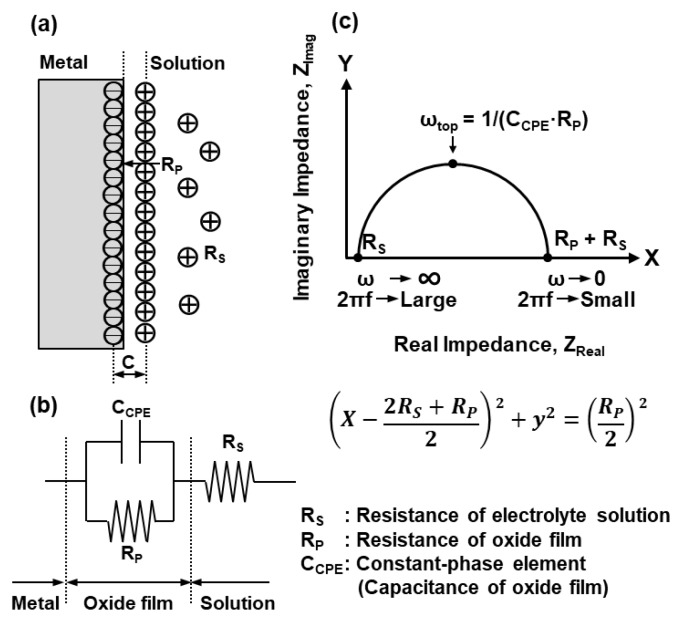
(**a**) Electrochemical model of oxide film, (**b**) equivalent circuit model of oxide film used to fit the impedance data, and (**c**) schematic illustration of Nyquist plot.

**Figure 2 materials-12-03466-f002:**
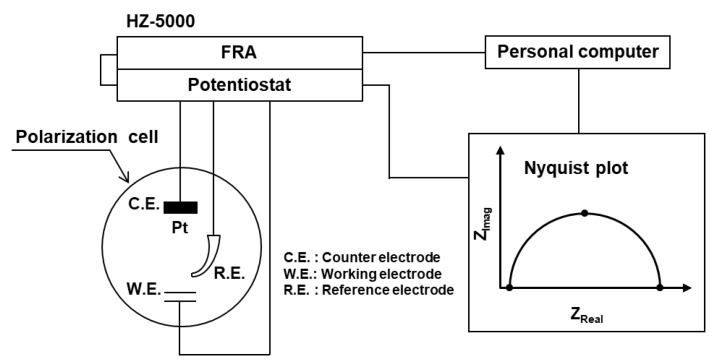
Experimental apparatus used for electrochemical impedance measurement.

**Figure 3 materials-12-03466-f003:**
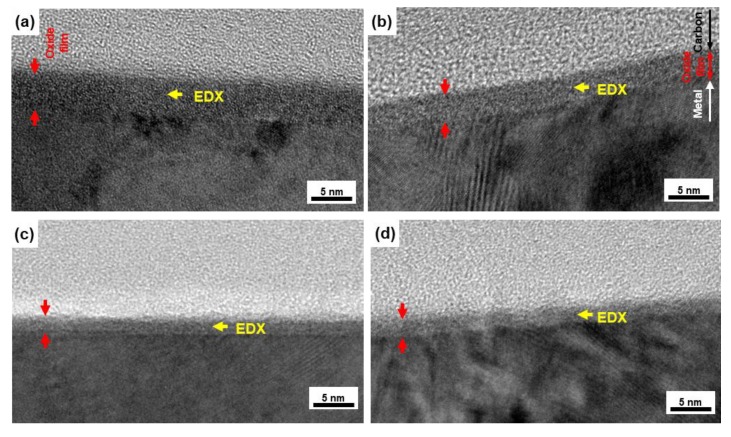
FE-TEM images of oxide films formed on (**a**) Ti–15Zr–4Nb–1Ta, (**b**) Ti–6Al–4V, (**c**) Co–28Cr–6Mo alloy, and (**d**) high-N stainless steel surfaces by anodic polarization up to 0 V vs. SCE in 0.9% NaCl at 37 °C.

**Figure 4 materials-12-03466-f004:**
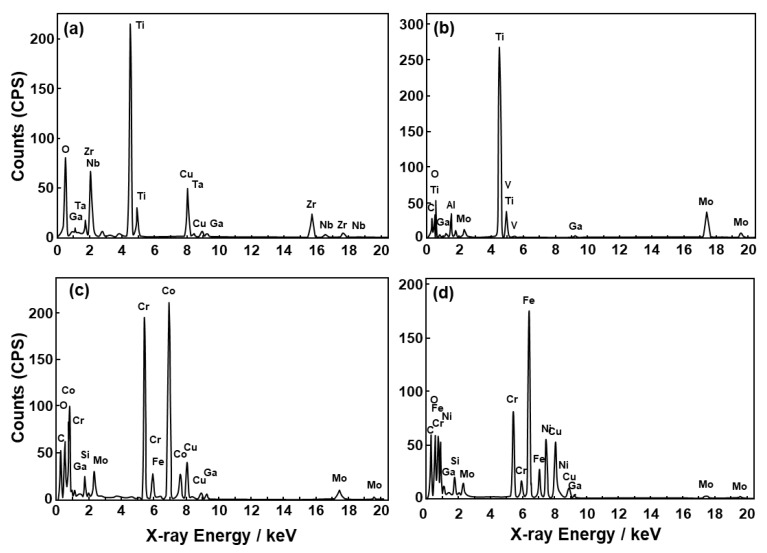
EDX spectra of oxide films formed on (**a**) Ti–15Zr–4Nb–1Ta, (**b**) Ti–6Al–4V, (**c**) Co–28Cr–6Mo alloy, and (**d**) high-N stainless steel at the positions shown in Figure 3.

**Figure 5 materials-12-03466-f005:**
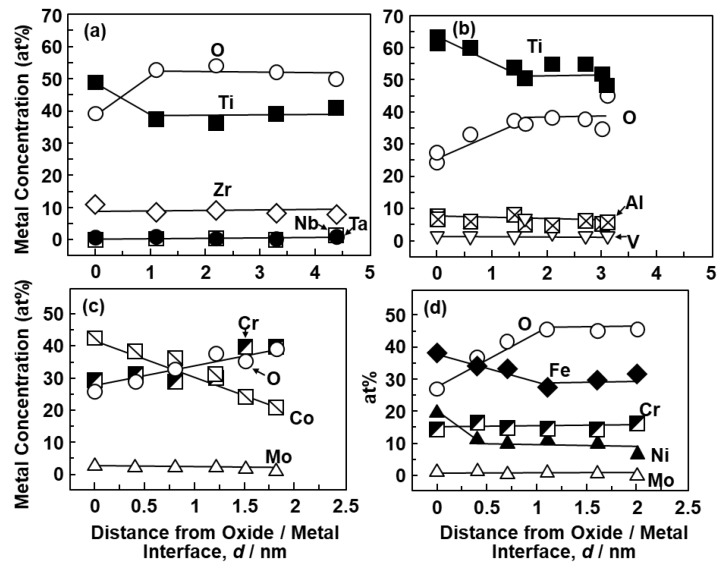
Changes in metal concentrations (at%) in oxide films formed on (**a**) Ti–15Zr–4Nb–1Ta, (**b**) Ti–6Al–4V, (**c**) Co–28Cr–6Mo, and (**d**) high-N stainless steel from oxide/metal interface to oxide film surface.

**Figure 6 materials-12-03466-f006:**
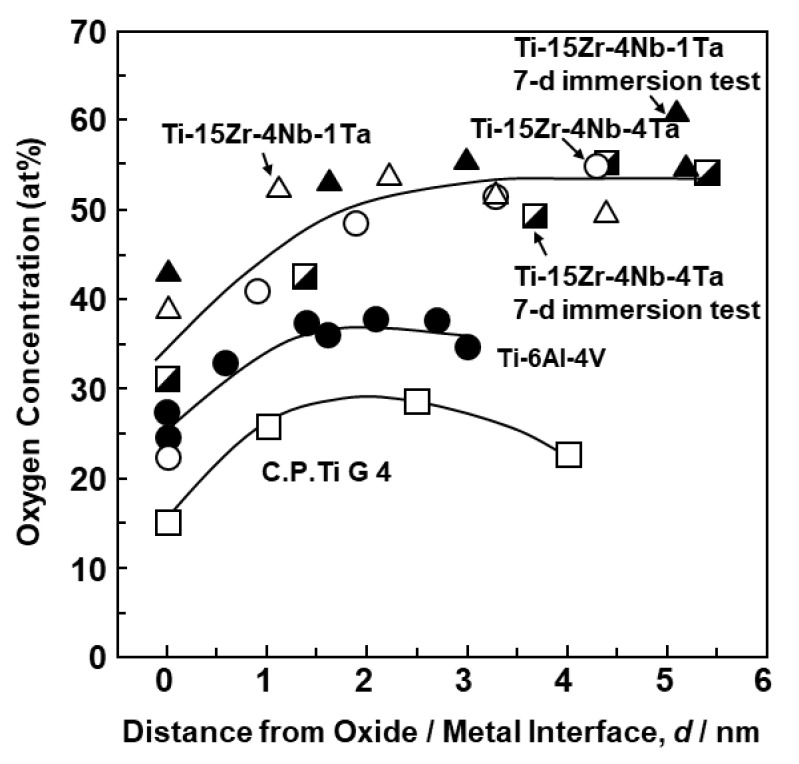
Changes in oxygen (O) concentration (at%) in oxide films formed on Ti–15Zr–4Nb–4Ta, Ti–15Zr–4Nb–1Ta, Ti–6Al–4V, and C.P.-Ti grade 4 surfaces by anodic polarization up to 0 mV vs. SCE or in 7-d immersion test for Ti–15Zr–4Nb–1Ta and Ti–15Zr–4Nb–4Ta alloys in 0.9% NaCl at 37 °C.

**Figure 7 materials-12-03466-f007:**
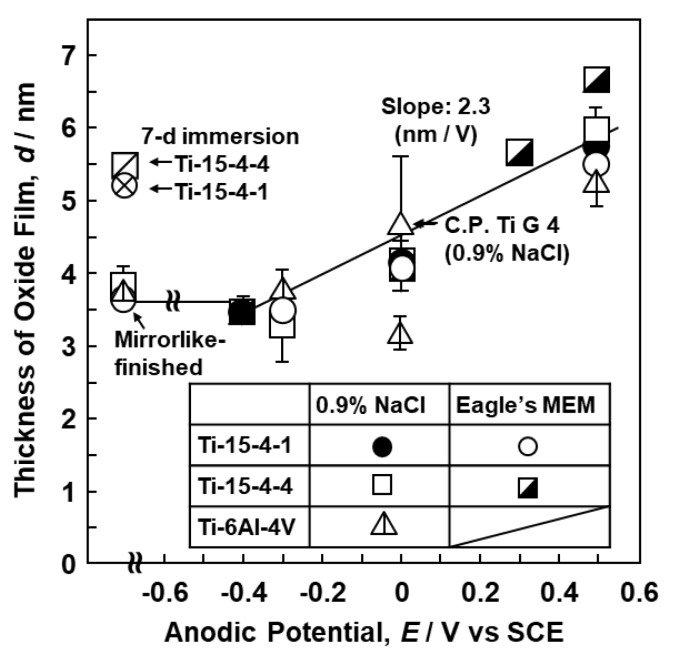
Effect of anodic polarization on thickness of oxide films formed by anodic polarization on surfaces of Ti–15Zr–4Nb–4Ta, Ti–15Zr–4Nb–1Ta, and Ti–6Al–4V alloys and C.P.-Ti G-4 in 0.9% NaCl or Eagle’s MEM solution at 37 °C.

**Figure 8 materials-12-03466-f008:**
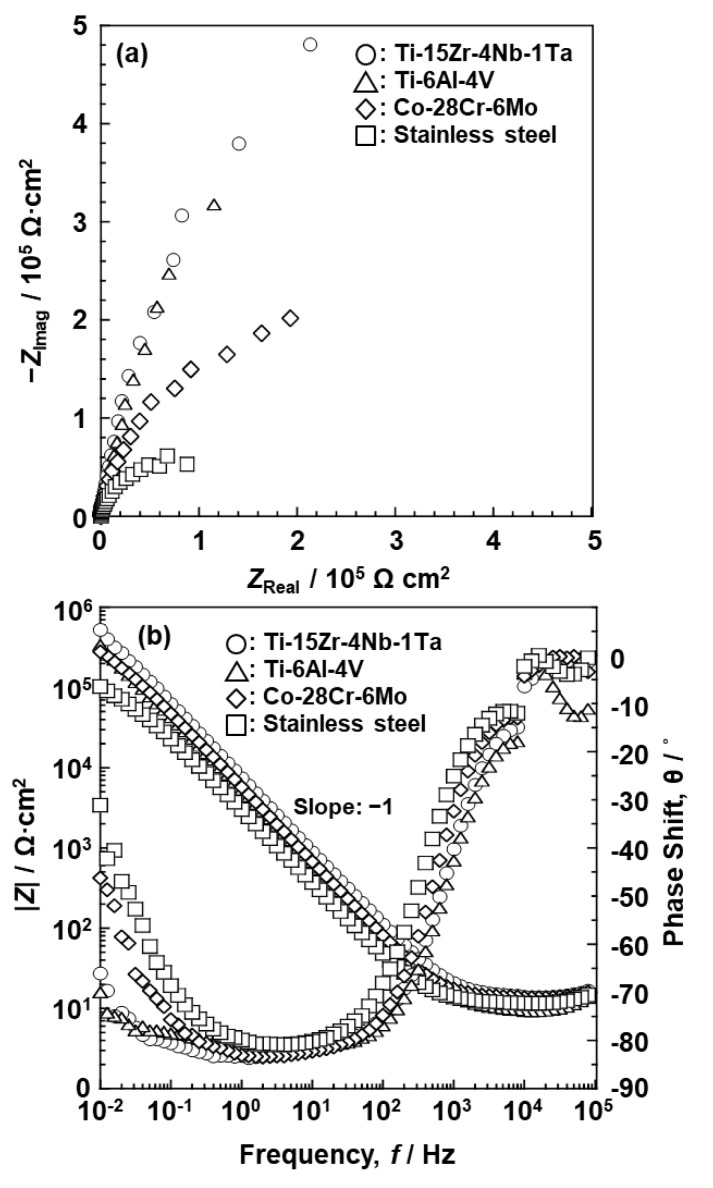
(**a**) Nyquist and (**b**) Bode plots of oxide films on Ti–15Zr–4Nb–1Ta, Ti–6Al–4V, Co–28Cr–6Mo alloy, and high-N stainless steel surfaces after anodic polarization up to 0 V vs. SCE in 0.9% NaCl solution at 37 °C.

**Figure 9 materials-12-03466-f009:**
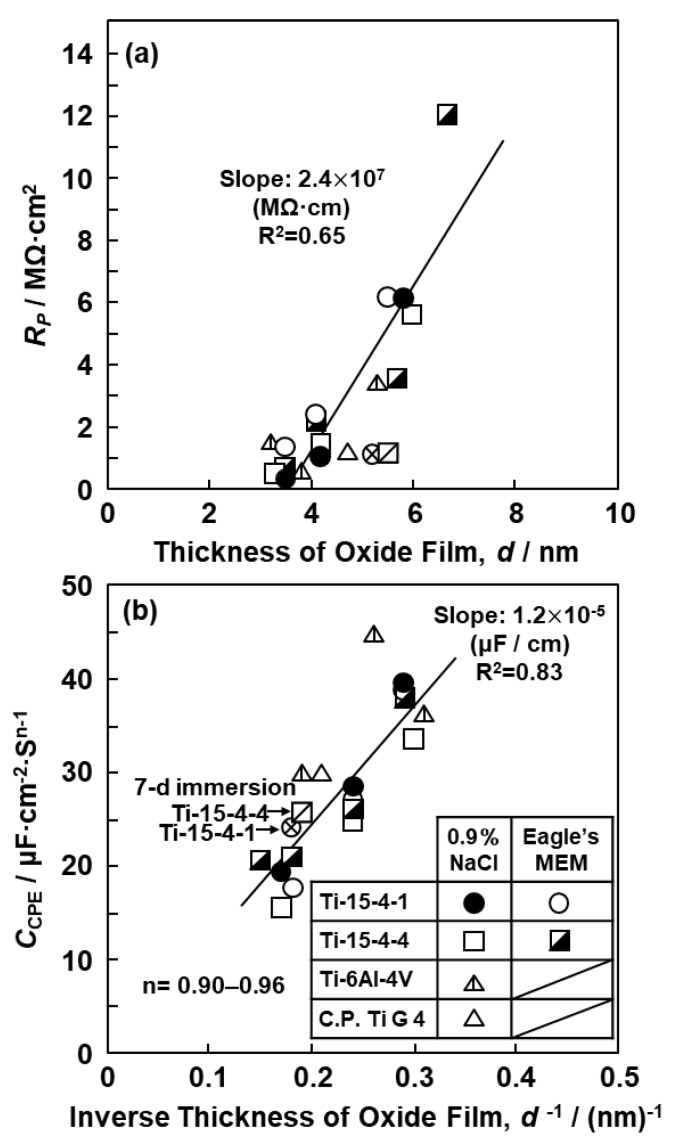
(**a**) Plot of oxide film resistance (R_P_) of Ti–15Zr–4Nb–1Ta, Ti–15Zr–4Nb–4Ta, Ti–6Al–4V, and C.P.-Ti G-4 vs. the oxide film thickness (d) obtained by FE-TEM. (**b**) Plot of capacitance (C_CPE_) of oxide films vs. 1/d obtained by anodic polarization test or by 7-d immersion test in 0.9% NaCl and Eagle’s MEM solutions at 37 °C.

**Figure 10 materials-12-03466-f010:**
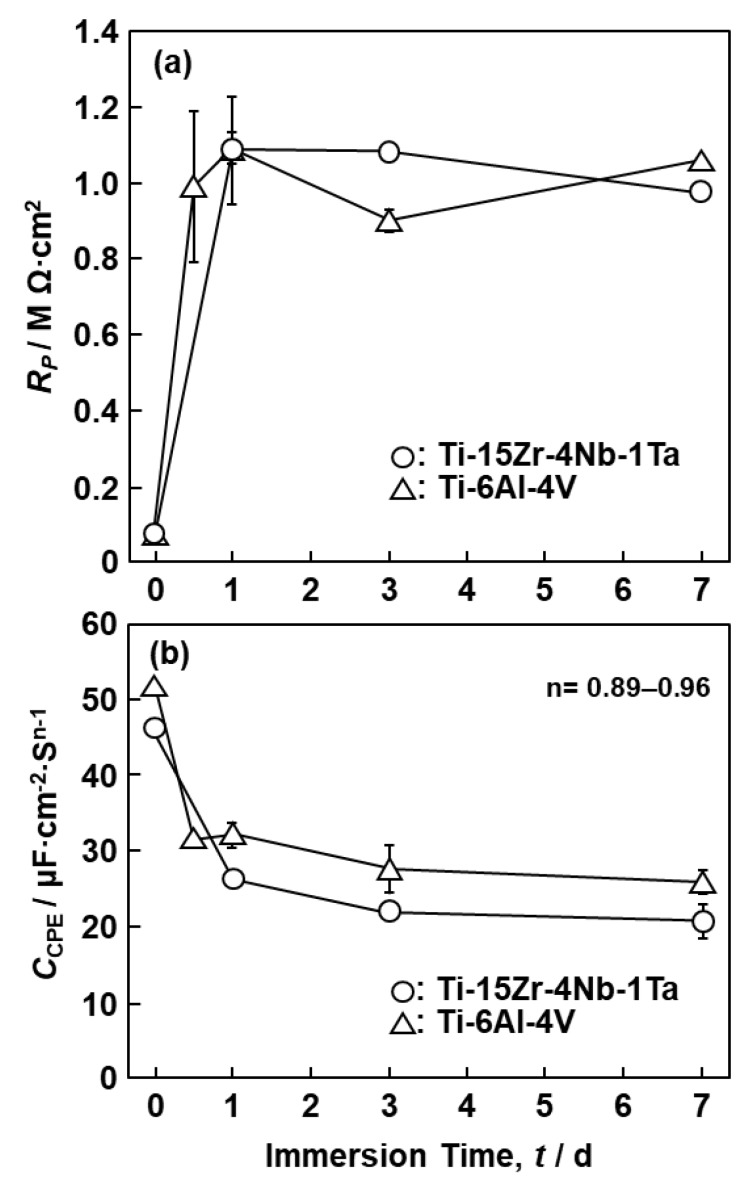
(**a**) Resistance of oxide film (R_P_) and (**b**) constant-phase element (capacitance) of oxide film (C_CPE_) formed on surfaces of Ti–15Zr–4Nb–1Ta and Ti–6Al–4V alloys after different immersion times in 0.9% NaCl solution at 37 °C.

**Figure 11 materials-12-03466-f011:**
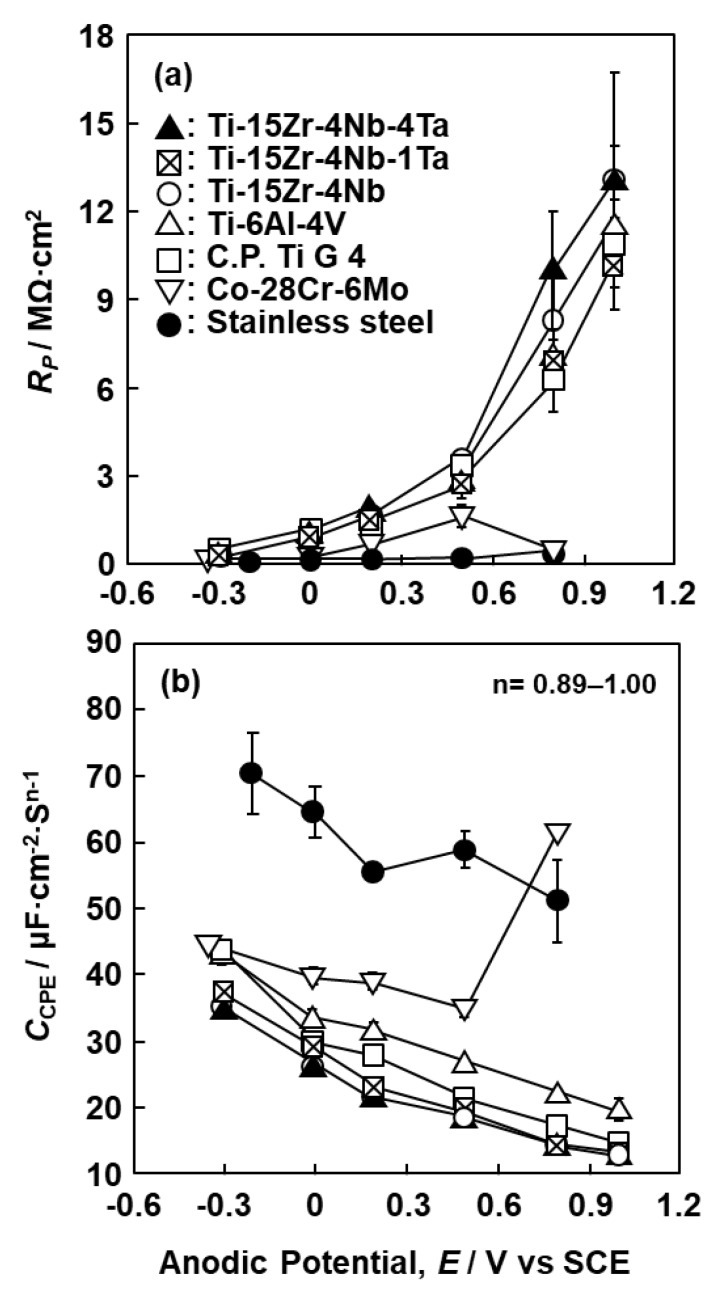
(**a**) Resistance of oxide film (R_P_) and (**b**) capacitance of oxide film (C_CPE_) formed on surfaces of Ti–15Zr–4Nb–(0 to 4)Ta and Ti–6Al–4V alloys after anodic polarization at various potentials in 0.9% NaCl solution at 37 °C.

**Figure 12 materials-12-03466-f012:**
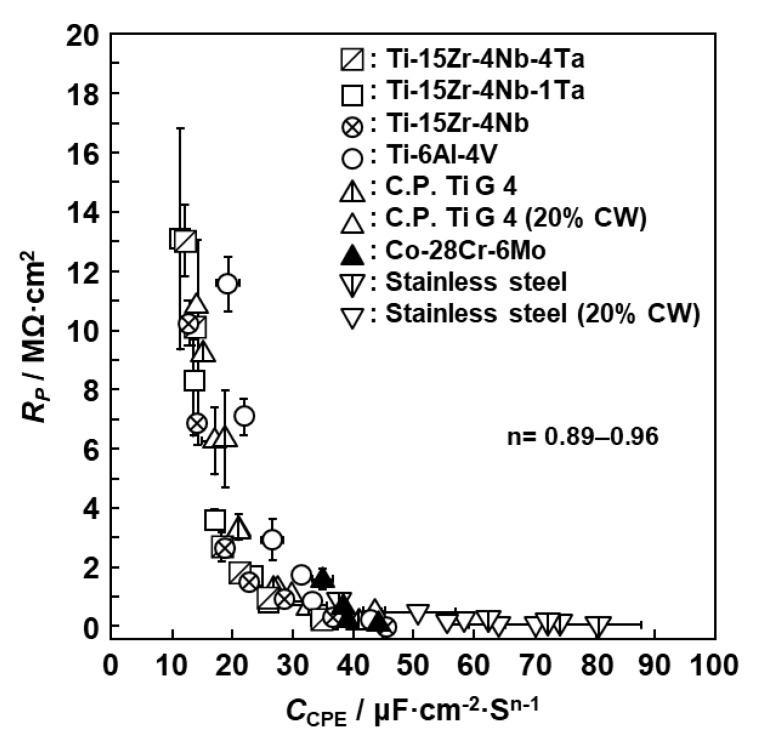
Relationship between resistance (R_P_) and capacitance (C_CPE_) of oxide films formed on various metals in the anodic polarization test up to 1 V vs. SCE in 0.9% NaCl at 37 °C after polishing with 1000-grit emery paper.

**Figure 13 materials-12-03466-f013:**
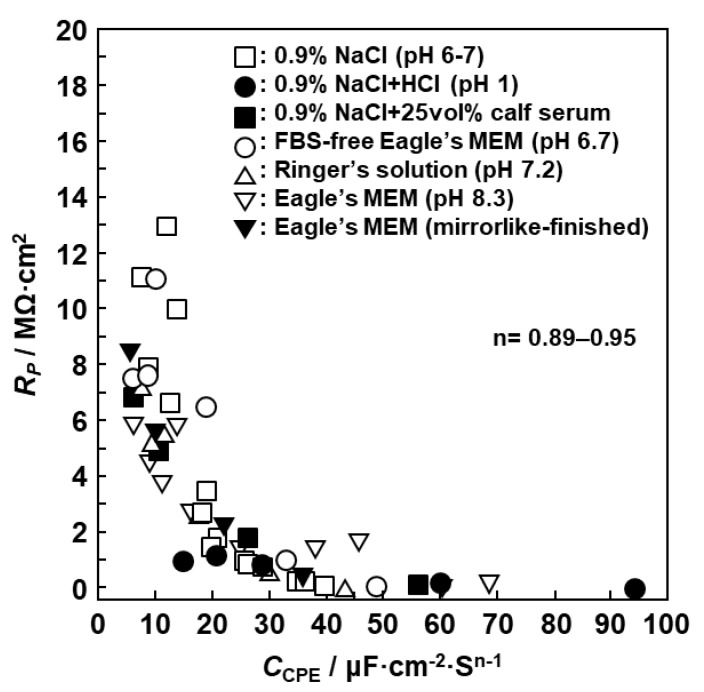
Effects of biological solutions and pH on R_P_ and C_CPE_ diagram of oxide films formed on 1000-grit-polished Ti–15Zr–4Nb–4Ta alloy.

**Table 1 materials-12-03466-t001:** Chemical compositions (mass%) of the Ti–15Zr–4Nb–4Ta, Ti–15Zr–4Nb–1Ta, and Ta-free Ti–15Zr–4Nb alloys, C.P.-Ti Grade 4, Ti–6Al–4V, high-N stainless steel, and Co–28Cr–6Mo alloy used.

Alloy	Zr	Nb	Ta	Al	V	Pd	Fe	O	N	H	C	Ti
Ti-15-4-4	16.55	4.0	3.9	−	−	< 0.01	0.04	0.28	0.09	0.0012	0.007	Bal.
Ti-15-4-1	17.24	3.97	1.67	−	−	0.02	0.036	0.29	0.096	0.0027	0.011	Bal.
Ti-15-4	17.1	4.2	−	−	−	< 0.01	0.026	0.28	< 0.01	< 0.001	< 0.01	Bal.
C.P. Ti G-4	−	−	−	−	−	−	0.197	0.275	0.003	0.0069	0.011	Bal.
Ti-6-4	−	−	−	6.4	4.4	−	0.10	0.07	0.02	0.0027	0.025	Bal.
	**Cr**	**Mo**	**Ni**	**Cu**	**C**	**N**	**Mn**	**Si**	**P**	**Be**	**Fe**	**Co**
Stainless steel	18.33	3.48	14.56	< 0.01	0.021	0.157	1.77	0.79	0.016.	0.0014	Bal.	−
Co-Cr-Mo	27.82	6.23	0.51	−	0.26	−	0.6	0.6	−	−	0.68	Bal.

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
