# Peer review of "Characterization of Oxide Film of Implantable Metals by Electrochemical Impedance Spectroscopy"

_materials, 2019, doi:10.3390/ma12213466_

Round 1
Reviewer 1 Report
The manuscript’s content is important for the scientific community and practical application of human-related activities. The author obtained interesting results. The manuscript’s text and results presented there are comments and suggestions.
It is not clear why the author is limited by the temperature of the experiments at 37 Celsius degrees, because the temperature can be higher!!!
In Fig. 4, the author presents the results of an EDX analysis at a voltage up to10 keV, the range should be increased, otherwise it turns out that the target element chemical composition (Mo) looks like an impurity (4c, and d).
The chemical composition and concentration of elements should be given in uniform units: in weight % (mass % or wt %) or atomic % (at. %), or both.
@Good luck@
Author Response
Reviewer 1
Thank you for the peer review of the manuscript, which has been revised in accordance with your comments. The modifications are as follows:
(1) I have added that the temperature of test solutions is controlled within the range of 37±1 °C.
(2) The EDX measurement range in Fig. 4 has been corrected to a wider range of voltages.
(3) I have modified the units (mass% or at%) of the chemical compositions and the concentration of each metal element.
Reviewer 2 Report
The proposed paper presents Electrochemical Impedance Spectroscopy (EIS) measurements of oxide films formed on metals. The measurements have been carried out on different metals/conditions and the results compared. I think the authors must address the following issues:
In the paper EIS measurements have been carried out on different metals/conditions, the corresponding electrical parameters from the equivalent circuit calculated and the results compared. However, the objective of the work is not very clear. At the end of the introduction there is the statement “The impedance characteristics obtained in this study are useful for the development of biomaterials with excellent biocompatibility” and at the end of the “Results and Discussion” section (lines 329-330) there is the sentence “The results suggest that Rox should be used as a parameter for assessing biological safety”. The objective of the work must be discussed in deeper details. For example, what electrical parameters are the most important and why? From these results, what of the investigated materials/conditions are better for the biocompatibility of implanted metals and why? Section 2.3 features different inaccuracies and must be revised. For example, at line 116 there is a reference to eq. (4) but this equation is not referred in the text but only in Fig. 1. At line 116: “alternating impedance” should be replaced with “complex impedance”. Line 118: wtop=1/C/Rox should be wtop=1/(CRox). At lines 118-121 there is the sentence “In the case of anodic oxides, strictly speaking, a multilayer model including a Helmholtz double layer is required. In this work, however, the simple (single-oxide-layer) model was used ….”. Why using a single-layer model if a multi-layer model is required? In Fig. 1 the resistance of oxide film is expressed as Rp in the equation instead as Rox. In Fig. 1 the equivalent electrical circuit features an ideal capacitance to model the capacitance of the oxide film. However in the fitting of the measured data a constant phase element was used. In Figure 9 the electrical parameters Rox and Ccpe have been plotted vs d and 1/d respectively. However the obtained coefficient of determination R^2 is low (in particular in the case of Rox). What are the reasons for the deviation from the linear relation presented in eqs. (1) and (2)?Author Response
Reviewer 2
Thank you for the peer review of the manuscript, which has been revised in accordance with your comments. The modifications are as follows:
(1) The purpose of the research has been added as follows.
The purpose of this study is to establish RP and C diagrams of implantable metals and, in particular, to estimate the wide range of RP and C values for Ti materials, stainless steel, and Co-Cr-Mo alloy. Another aim is to obtain the electrochemical data of typical implantable metals under the same conditions. The RP and C diagrams obtained in this study are useful for the development of new metallic materials with excellent corrosion resistance. For a metallic material having higher RP, the quantity of ions that pass through the surface oxide film is reduced, and such a material exhibits higher corrosion resistance. The RP and CCPE diagrams obtained in this study are useful for the development of new metallic materials with excellent corrosion resistance.(2) I have modified Section 2.3 as follows.
Eq. 4 has been added in the text and Fig. 1 has also been modified in accordance with your comments. “alternating impedance” was replaced with “complex impedance”. ωtop=1/C/Rox has been replaced with ωtop =1/(CCPE·Rox), and Rox has been replaced with RP.
(3) The reason why the linearity of Fig. 9(a) is low is newly described as follows.
As clearly shown in Fig. 9(a), RP did not lie on a straight line passing through the origin, which is different from the result indicated in Eq. (2), and R2 (0.65) of the slope was lower. This may have been caused by the effect of thin film that initially formed on the metal surface during mirrorlike polishing. We would like to investigate the cause of this difference in the future and improve the electrochemical model of the oxide film.
(4) The following sentences have been deleted because I wanted to consider their contents in my next study.
In the case of anodic oxides, strictly speaking, a multilayer model including a Helmholtz double layer is required [18]. In this work, however, the simple (single-oxide-layer) model was used to analyze the impedance data to construct the electrochemical data of typical implantable metals under the same conditions.
Round 2
Reviewer 2 Report
The paper has been revised according to the reviewers’ comments. I think it can be published if the authors address the following minor revisions:
The authors have changed the symbol for the oxide resistance from Rox to Rp. However the old symbol Rox is still present at the following lines in the manuscript: line 18, line 53, line 59, line 252, line 361. Line 64: “obtain” and not “obtaine”. Line 290: there is a symbol epsilonT that I think is epsilonr
Author Response
Reviewer 2
Thank you for the peer review of the manuscript, which has been revised all in accordance with your comments.